# The Role of Music Therapy with Infants with Perinatal Brain Injury

**DOI:** 10.3390/brainsci12050578

**Published:** 2022-04-29

**Authors:** Kirsty Ormston, Rachel Howard, Katie Gallagher, Subhabrata Mitra, Arthur Jaschke

**Affiliations:** 1Noah’s Ark Children’s Hospice, University College Hospital London, London EN5 4NP, UK; 2Institute for Women’s Health, University College, London WC1E 6HU, UK; rachel.howard.20@ucl.ac.uk (R.H.); katie.gallagher@ucl.ac.uk (K.G.); subhabrata.mitra.13@ucl.ac.uk (S.M.); 3University Medical Centre Groningen, ArtEZ University of the Arts, 6812 CE Arnhem, The Netherlands; A.jaschke@artez.nl

**Keywords:** neonate, hypoxic ischemic encephalopathy, perinatal brain injury, music therapy, auditory stimulation, vocal stimulation

## Abstract

Perinatal brain injury occurs in 5.14/1000 live births in England. A significant proportion of these injuries result from hypoxic ischaemic encephalopathy (HIE) in term infants and intracranial haemorrhage (IVH) or periventricular leukomalacia (PVL) in preterm infants. Standardised care necessitates minimal handling from parents and professionals to reduce the progression of injury. This can potentially increase parental stress through the physical inability to bond with their baby. Recent research highlights the ability of music therapy (MT) to empower parental bonding without handling, through sharing culturally informed personal music with their infant. This review therefore aimed to systematically evaluate the use of MT with infants diagnosed with perinatal brain injury in a neonatal intensive care unit (NICU). Search terms were combined into three categories (audio stimulation (MT), population (neonates) and condition (brain injury), and eight electronic databases were used to identify relevant studies following PRISMA guidelines. Eleven studies using music or vocal stimulation with infants diagnosed with perinatal brain injury were identified and quality assessed using Cochrane ROB2, the ROBINSI Tool and the Newcastle Ottawa Scale. Studies used either voice as live (n = 6) or pre-recorded (n = 3) interventions or pre-recorded instrumental music (n = 2). Studies had two primary areas of focus: developmental outcomes and physiological effects. Results suggested the use of music interventions led to a reduction of infants’ pain scores during procedures and cardiorespiratory events, improved feeding ability (increase oral feeding rate, volume intake and feeds per day) and resulted in larger amygdala volumes than control groups. Additionally, MT intervention on the unit supported long-term hospitalised infants in the acquisition of developmental milestones. Vocal soothing was perceived to be an accessible intervention for parents. However, infants with PVL showed signs of stress in complex interventions, which also potentially resulted in an increase in maternal anxiety in one study. MT with infants diagnosed with perinatal brain injury can have positive effects on infants’ behavioural and neurological parameters and support parental involvement in their infants’ developmental care. Further feasibility studies are required using MT to determine appropriate outcome measures for infants and the support required for parents to allow future comparison in large-scale randomised control trials.

## 1. Introduction

Perinatal brain injury is diagnosed in 5.14 in every 1000 live births in England [1]. These infants are admitted to the neonatal intensive care unit (NICU) shortly after birth. This is often an unexpected experience for parents, interrupting the relationship that has been building with their infant during pregnancy [2]. This interruption combined with the parents’ heightened awareness of the possibility of loss can result in parents experiencing symptoms of trauma, impacting their relationships with others, particularly their baby [3,4].

There are several causes of perinatal brain injury, including neonatal encephalopathy (NE). One of the causes of NE, hypoxic ischaemic encephalopathy (HIE), affects approximately 3–4 in every 1000 live births [5]. Approximately 50% of infants diagnosed with HIE who receive therapeutic hypothermia have outcomes that include death or long-term disability including epilepsy, cerebral palsy and cognitive impairment [6]. In pre-term infants, perinatal brain injury can result following intracranial haemorrhage (IVH) and periventricular leukomalacia (PVL), the incidences of which are increased when infants are born <27 weeks and/or with a very low birth weight [7]. Both IVH and PVL can result in an increased risk of cognitive impairment and cerebral palsy [8]. A further aetiology is perinatal stroke, occurring in approximately 1:1600–1:5000 live births [9]. The severity of neonatal encephalopathy or detection of seizures related to injury are revealed through diagnostic and monitoring techniques such as neuroimaging via magnetic resonance imaging (MRI) and cranial ultrasound and monitoring with the use of electroencephalogram (EEG) and near-infrared spectroscopy (NIRS) [10]. Both therapeutic hypothermia in term infants and the initial care bundle for the prevention of IVH in preterm infants demand minimal handling from parents and neonatal healthcare professionals [11]. This minimal handling has been shown to increase stress for parents [12]; the resulting stress and anxiety can profoundly impact upon parent–infant bonding [13]. When bonding is challenged, parents’ efficiency in interpreting their infant’s cues can be adversely affected, thus impacting upon their interaction with their baby, parental mental health and potentially the subsequent longer term neurodevelopment of the infant [14,15].

Music therapy (MT) as a family intervention in neonatal settings has been found to potentially reduce feelings of parental stress whilst empowering parents to engage in the developmental care of their baby [15,16,17]. It has also been shown to be a supportive intervention for parents experiencing anticipatory grief [16]. Music therapy is an established clinical psychological intervention provided by a registered therapist that uses the qualities of music to build a therapeutic relationship between the therapist and client [17]. This therapeutic relationship aims to support the physical, emotional, cognitive and social needs of people of all ages [17]. Increasingly, MT is being implemented in neonatal units globally, following the developments of specially tailored approaches for working within this environment, considering the relationship between parents, infants and the environment [18,19,20]. These approaches include the provision of music in the form of live infant-directed singing provided by trained therapists or parents, the sharing of culturally valued songs and pre-recorded lullabies provided using pacifier-activated music players. Music therapy can be used in combination with other developmentally appropriate interventions in a neonatal unit, such as kangaroo care or multi-model stimulation bundles [21,22]. Whilst the evidence base to support the use of MT in neonatal units is still growing, the limited research undertaken with preterm infants highlights that MT can improve an infant’s physiological parameters, behaviour states, weight gain and feeding ability [18,23]. Research has also demonstrated beneficial effects for preterm infants’ functional brain activity and connectivity [24] and cerebral oxygenation [25] due to MT’s ability to create social connectivity [26] and its potential to provide a stimulating or sedative effect as appropriate.

Currently, newborn infants with perinatal brain injury are often excluded from studies, potentially due to their fragility and complexity as a research population as well as the increased likeliness of parents experiencing depression, distress and anxiety during this emotionally challenging time [27]. However, paediatric research has identified that outcomes of MT in children with acquired and traumatic brain injury have been improved when involving the parent or caregiver [28]. Due to the complexity of developmental support required for infants with perinatal brain injury, it is likely that outcomes of interventions will be higher with parental involvement. This could potentially be achieved through the psychotherapeutic approach that a music therapist can provide. Therapists are able to highlight cues, model interactions and build confidence in parent’s own parenting abilities [24]. It may also be possible for therapists to provide consultation post-discharge or ideas for continued developmental support. In order to fully understand what this support would entail for both parents and their infant, further investigation of the potential impact of MT with infants with perinatal brain injury is required.

### Aim

This review aims to systematically evaluate existing evidence exploring the use of MT with infants diagnosed with perinatal brain injury. It assesses (a) the effects of auditory stimulation in the form of live performed and pre-recorded music as well as vocal soothing on neonatal physiological and neurological parameters with perinatal brain injury compared to standard neonatal intensive care and (b) the effect of MT on parental experience following the admission of their infant to a neonatal unit for treatment following diagnosis of perinatal brain injury. We then discuss whether MT is potentially an appropriate intervention for infants with perinatal brain injury.

## 2. Methodology

### 2.1. Information Sources and Search Terms

Published articles were identified following guidance from the Preferred Items for Systematic Reviews and Meta-Analysis (PRISMA) statement [29]. Comprehensives searches of eight databases were conducted by a specialist librarian: MEDLINE, Embase, AMED, Psych INFO, CINAHL PLUs, the Cochrane Library, Web of Science and PubMed. All studies from inception to December 2021 were included without restrictions on language. Search terms were combined into three categories: auditory stimulation (music therapy), population (neonates) and condition (brain injury). Three sets of heading terms were used for searches:Music therapy, acoustic stimulation, auditory stimulation, music intervention, music, voice, song, sound, vocal, singing, womb sounds, heartbeatNeonatal, neonate, newborn, new-born, preterm, pre-term, extreme preterm, very preterm, moderate to late preterm, prematureBrain injury, seizure, stoke, neonatal encephalopathy, traumatic brain injury, Hypoxic ischemic encephalopathy, HIE, intrapartum asphyxia, foetal brain injury or fetal, IVH, intraventricular haemorrhage, PVL, periventricular leukomalacia

Heading terms were then linked by “OR” and searched with other sets using “AND” as shown in Table 1. A second search of the databases was conducted 2 months after the initial searches by the primary researcher, with a third search prior to publication to confirm the search result and ensure the searches represented the most current data available.

### 2.2. Selection Process

Publications were included in the review if they presented data that used MT as an intervention with neonates with brain injury. This included all infants admitted to the neonatal unit, preterm or full term, who experienced brain injury in the perinatal period.

The intervention included any musical or vocal stimulation that was provided to both the infant alone and with the parents present. Studies where the musical intervention was combined with other interventions such as feeding and skin-to-skin were also included. Due to MT being an emerging field, the intervention could be provided by any healthcare professional or parent but had to (a) last a minimum of five minutes (b) be provided for a minimum of two sessions. Digital signals and white noise were excluded as forms of auditory stimulation as they do not contain inflection or speech-like rhythmic patterns. Those which included the use of parental/caregiver voice were included due to the emotive, reflective quality and prosody of parent–infant interaction. Only human studies were included for review.

The first two authors (KO and RH) independently screened publications from titles and abstracts collated based on the inclusion criteria (Table 2). Publications grouped as “maybe” were then discussed to determine inclusion/exclusion. Full text papers of any titles considered to meet the criteria by either author were obtained for further screening. Any studies that did not meet the inclusion criteria were excluded. One article was a review of a paper, and therefore the original paper was obtained; however, it was subsequently excluded as it did not meet the inclusion criteria.

### 2.3. Quality Appraisal

The quality of each included study was assessed using the Cochrane risk of bias tool (RoB2) for randomized controlled trials and the ROBINSI Tool for non-randomised studies [30] prior to data extraction. The quality assessment included six aspects of risk of bias. Aspects were rated as low, high or some concerns based on the algorithm for suggested judgement provided in the RoB2 crib sheet [31]. The Cohort study was quality assessed using the Newcastle–Ottawa quality assessment scale, awarding stars based on the coding scale [32] (see Appendix A).

### 2.4. Subgroup Analysis

Outcomes of the form intervention were explored further through subgroups of pre-recorded music and live vocal stimulation to evaluate any changes in effectiveness for the population as a result of this. These effects could then determine the safest recommended protocol for working with this population in future studies. The postmenstrual age of infants was analysed to determine any impact upon the effectiveness of the intervention upon infant physiological and neurological outcomes.

## 3. Results

This initial search identified 8618 articles. After screening abstracts and titles, 33 articles were selected to include the use of MT in the form of vocal stimulation or pre-recorded music as an intervention with neonates diagnosed with brain injury on the neonatal unit. After reviewing full text publications, 11 works were identified for final review. The PRISMA chart detailing the search results and screening process is presented in Figure 1. Basic characteristics and brief details of the studies are presented in Table 3. Eleven studies were published between 1999 and 2021. Studies had two primary areas of focus: developmental outcomes and physiological effects on infants, focusing on oxygen saturation and heart rate.

### 3.1. Forms of Intervention

Interventions in studies included live sung female voice (n = 2), recorded maternal voice (n = 3), recorded instrumental music (n = 2) and vocal soothing as part of a multimodal bundle (n = 4). There was no evidence found in this review that the form of music intervention impacted upon either the infant or parental experience; however, only one study discussed the value of trained professionals in supporting the provision of music to avoid the potential dangers of over-stimulation. Similarly, the impact of personalising the intervention to the unique identities that each individual family brings was not mentioned. Of the 11 studies reviewed, 9 provided interventions using a female or maternal voice (n = 3 pre-recorded) in contrast to pre-recorded instrumental music (n = 2). None of the identified studies directly compared live and pre-recorded music. The intervention was provided by a qualified music therapist in three of the studies, resulting in the majority of the results being an outcome of exposure to music rather than having a psychotherapeutic approach. Four studies provided vocal stimulation and soothing as part of a multisensory bundle (n = 2 caregiver led, n = 2 healthcare professional administered); these were included due to the interpersonal connectivity of caregiver–infant interaction and the song-like manner that infant interaction is often carried out in.

Studies using recorded maternal voice ensured that guidelines on the use of recorded music were followed to ensure that infants were not exposed to inappropriate decibel levels [33,34]. Importantly, in the studies, pre-recorded music was used in a controlled environment where the infant was being observed, and music could be stopped if it had an adverse effect. One study provided further variation for the use of pre-recorded music by providing a variety of music that could be chosen. Selection was based on the observed infant’s state, showing the importance of music being infant-led in order to be supportive. Whilst most articles reviewed commented on the behavioural state of the infant before interventions, four papers did not specify the state of the infant at the time of intervention.

### 3.2. Immediate Impact on Newborn Physiological Parameters

Four studies considered the effect of pre-recorded music on infant physiological parameters [35,36,37,38], and of these, three used a maternal voice. Infants included in the studies ranged between 31 and 34 weeks post-menstrual age at the start of the intervention. Chorna et al. explored the impact of receptive music upon the development of sucking ability in infants 34 weeks postmenstrual (46 experimental, 48 control) using the Pacifier Activated Music (PAM) player for 5 consecutive days. This system delivered music for 15 min on detection of a suck that met a pre-set threshold. The music delivered to the infant was a recording of lullabies sung by the infant’s mother with the support and guidance of a trained music therapist. The authors found a positive association between the use of PAM and an improvement in feeding skills in infants diagnosed with small IVH (grades I and II). A significant increase in oral feeding rate (*p* < 0.001), oral volume intake (*p* = 0.001), number of oral feeds per day (*p* < 0.001) and faster time to full (*p* = 0.04) was shown by infants receiving musical intervention; however, these results did not allow for the impact of the intervention on infants with brain injury to be considered separately. The authors commented that recording a mother’s voice increases maternal presence and therefore can be used for positive association with the infant. Two studies (n = 1 maternal voice, n = 1 instrumental music) documented positive effects of the music intervention on lowering staff-reported (doctor/nurse/researcher) infant pain-scores assessed using the Neonatal Pain Agitation and Sedation Scale (N-PASS) [39] and Premature Infant Pain Profile [40]. These studies provided pre-recorded music to infants of 33 and 34 weeks postmenstrual during painful procedures such as cannulation and line insertion [36,38]. The interventions differed in duration: Chirico et al. provided 2 sessions daily for 3 days, compared to Konar et al. who provided 4 sessions daily, which has been assumed to continue until 24 months corrected age. Chirico et al. documented the provision of music from 10 min before the procedure until 20 min after. Lower pain scores were recorded in both studies for the experimental groups (*p* < 0.01 and *p* = 0.0002 respectively), demonstrating the benefits of pre-recorded music when parents are unable to provide the intervention. Doheny et al. examined the impact of maternal sounds on cardiorespiratory events (defined as episodes of apnoea and bradycardia) of 14 infants between 26 and 32-weeks postmenstrual age during 24 h periods throughout the duration of their stay in the neonatal unit. The study provided four sessions administered by a healthcare professional, each lasting 30 min, with maternal sounds captured by a microphone and digital stethoscope through a micro audio system in the incubator. The authors reported significantly lower numbers of cardiorespiratory events (*p* = 0.03) and therefore an increased stability in infants ≥33 weeks postmenstrual age. The authors suggest this could be attributed to foetal development and the ability to distinguish moods and emotional qualities in sound established at around 33 weeks of gestation [41]. Despite encouraging findings, Doheny et al.’s study is limited by its small sample size and the inclusion of only a small population of infants with brain injury (7.1%), and it therefore cannot be used to generalise across the neonatal population.

The positive effect of voice on the physiological parameters of infants was reported in eight of the studies, supporting current evidence that the parental voice has a soothing and settling effect on infants [42,43]. Two of these studies, both from the USA, used a live voice [44,45]. Epstein et al. studied the provision of live maternal singing accompanied by a guitar or a second female voice for 35 infants, 32 ± 3.5 post-menstrual age, with severe brain injury (IVH grade 3 or 4 and/or PVL). The intervention was carried out 30 min after the completion of an infant feed (type of feed not specified) and in combination with kangaroo care. Sessions started with 10 min of skin to skin alone, followed by 20 min of musical intervention with skin to skin and ending with 10 min of recovery in skin to skin alone. Three sessions in total were provided, with the number of days between sessions varying. The authors found the addition of MT along with skin to skin resulted in a higher heart rate (*p* = 0.04) and higher infant behaviour state (*p* = 0.04) compared to skin to skin care alone. Results were documented by a research assistant who sat near the mother during the intervention.

White-Traut et al. (2003) undertook a controlled trial where the experimental group received a multi-sensory stimulation bundle, administered by a female research assistant, consisting of vocal interaction, eye contact, stroking and rocking twice daily from the point of recruitment until discharge. Thirty-seven premature infants born at 23–26 weeks with normal ultrasounds and born at 24–32 weeks with PVL and/or IVH were randomly assigned to control and experimental groups with results studied at 33–35 weeks of age. These were then analysed in groups categorised as (a) very low birth weight (VLBW) and IVH (b) PVL and PVL plus IVH. Over the three-week period, all infants in the control group showed a decrease in mean heart rate from baseline (−0.18 beats per minute (bpm) and −10.49 bpm respectively). However, in the experimental groups, infants with VLBW and IVH showed a 10.88 bpm decrease from baseline on completion of the intervention, whilst infants with PVL and PVL plus IVH showed a 12.67 bmp increase—a significant difference (*p* < 0.05). This increase in physiological parameters suggests the multiple forms of sensory stimulation were stressful for infants with PVL. This is potentially due to this population’s immature autonomic nervous system. White-Traut et al. (1999) undertook a similar study providing 15 min of auditory, tactile, visual and vestibular intervention twice a day, five days a week for four weeks, recruiting only infants diagnosed with PVL. This study notes a significant shift from sleep to alert states in experimental group infants with a prolonged alert state after the intervention. Infants were shown to suffer no injury as a result of the intervention and had an average hospital stay of nine days shorter than the control group. White-Traut et al. propose that the infant’s prolonged alert state creates a productive time for feeding post-intervention.

### 3.3. The Impact of MT on Infant Neurological Parameters

Three studies considered the impact of auditory stimulation on neurological parameters. Two of the studies provided auditory stimulation via pre-recorded instrumental music [38,46]. Konar et al.’s control trial provided 25–30 min of guitar music, four times daily to infants diagnosed with HIE and between 34.3 ± 2.1 weeks of gestation. Infants were provided with conventional management of birth asphyxia combined with music exposure by healthcare professionals on the unit followed by primary care givers after discharge. Whilst standard management of infants with birth asphyxia is mentioned, day of life at the time of the invention is not, and therefore it is unclear if the intervention was provided during therapeutic hypothermia. Infants in the experimental group had significantly higher neurodevelopmental quotient scores than those in the control group (*p* < 0.01) at 3, 6, 12, 18 and 24 months. Sa de Almeida et al. explored the impact of MT upon the maturation of brain regions involved in acoustic and or emotional processing in very preterm infants (n = 15 full term, n = 15 experimental preterm gestational age at birth <32 weeks, n = 15 control preterm). The study provided 8 min of recorded music, consisting of harp, punji (charming snake flute) and bells, selected according to the state of the infant, as assessed by clinical staff. The authors found that infants who had listened to music showed an increase in microstructural maturation in acoustic radiations, external capsule/claustrum/extreme capsule and uncinate fasciculus as well as significantly larger amygdala volumes than preterm controls (*p* = 0.006). These structures are all involved in acoustic/emotional processing. Whilst findings are encouraging, due to the small number of infants included with a diagnosis of perinatal brain injury (26.7% experimental, 6.7% control), the implications of the study for this population are limited. What this study does provide, however, is evidence to support the potential for musical intervention for mitigating later social–emotional difficulties associated with prematurity. Further investigation into the impact of preterm infants with brain injury is therefore required.

Nelson et al. explored the potential long-term impact of an auditory–tactile–visual–vestibular stimulation intervention on motor and cognitive skills, using the Bayley Scales of Infant Development [47], and mother–infant interaction in 37 infants with central nervous system injury or extreme prematurity (born at 23–26 weeks gestation). The intervention was provided at 33 weeks postmenstrual age and recommended to be continued until 12 months of age. Although the experimental group showed 23% (from 44–67%) fewer cerebral palsy diagnoses at 1 year of age, this was not found to be significant. When Bayley scores were compared between infants with PVL and extreme preterm infants without central nervous system injury, those without injury had significantly higher cognitive scores (*p* < 0.05). The authors suggest that PVL results in continued developmental delay despite intervention. The study reported a 30% loss to follow up, and therefore the results are limited; however, it should be noted that, whilst in the unit, there were few differences observed between the synchronicity of interactions of mother–infant dyads, based on the Dyadic Mutuality Code [48], of extreme preterm infants and those with PVL, suggesting a potential population group that can be closely compared when considering maternal interaction. Emery et al. provided eight music therapy sessions to long-term hospitalised infants in a neonatal unit (aged between 44–66 post-menstrual age) over a four-week period. Sessions were carried out with the infant out of the cot and included interactive musical play as well as reflective, created music. The study assessed the impact of session intensity (two sessions per week for four weeks versus four sessions in weeks 1 and 3 with no intervention in weeks 2 and 4) on infant’s acquisition of developmental milestones. Both intensities of intervention were shown to improve motor scores (*p* < 0.01) with the same degree of attainment. This demonstrates the benefit of music therapy in supporting long-term hospitalised infants in the attainment of developmental milestones and the potential for flexibility in the scheduling of sessions without inhibiting outcomes.

### 3.4. Impact of MT upon Parental Anxiety

Two studies evaluated the impact of the intervention on maternal stress and anxiety. Letzkus et al. started the use of auditory stimulation from the youngest age (from 23–25 weeks postmenstrual) of all studies. Once enrolled, mothers were taught the elements of a multi-stimulation bundle by a member of the research team familiar with the family nurture intervention by which the bundle is informed. Elements of the bundle were advised for different ages of infants, with the addition of massage being approved by a physical therapist before being implemented. Length of each interaction was based on maternal observation of their infant’s tolerance and engagement with the auditory stimulation element consisting of mothers expressing their feelings to their infants in their native language. In addition to support from a research assistant, mothers were provided with an intervention flip book for reminders on how to carry out the invention. The intervention was aimed to be provided 5/7 days a week for the duration of the infant’s NICU stay (median of 6 weeks) and was recorded in a diary by mothers. Comforting touch and scent exchange were reported to be feasible for parents to achieve each day as they both achieved the target performance of 5 days per week. Maternal stress, anxiety and depression were assessed through two self-assessed questionnaires—one prior to intervention and one at discharge. Letzkus et al. found a significant decrease in maternal stress (*p* = 0.02), identified through a reduction in parent-reported scores on the neonatal parental stressor scale (PSS-NICU [49]). This decrease was seen across all three components of the scoring system: visual stimuli and sounds, behaviour and appearance and parental relationship with infant. Interestingly, however, Epstein et al. reported an increase in maternal anxiety after MT combined with kangaroo care compared to kangaroo care alone [44]. This was measured through a self-reported questionnaire (state–trait anxiety inventory, STAI [50]) filled out by mothers after each intervention. As previously stated, Epstein’s study focused on the impact of maternal singing accompanied by a female voice or guitar during skin to skin on infants with severe brain injury, recording significantly higher heart rate (*p* = 0.04) and higher infant behaviour state (*p* = 0.04) for infants in the experimental group compared to skin to skin care alone. These studies support current research that the perceived stability of the infant contributes to maternal anxiety [51]. The difference in the way the studies were carried out may have impacted the parent’s evaluation of the study. In Letzkus et al.’s study, mothers were supported initially with the intervention and carrying out stimulation before then being left in control of when each element of the intervention was offered and when to move onto the next phase of the bundle; there was no additional observation of the mothers. Additional reminders of methods for carrying out the intervention were available in a flip book provided. In Epstein et al.’s study, however, mothers had no choice but to sing in the presence of a music therapist at specified times. Mothers may have felt that their singing would be criticised or judged by a professional musician or perceived pressure due to the specific nature of the study, thus potentially increasing their anxiety. 

## 4. Discussion

The aim of this review was to systematically evaluate evidence exploring the use of MT with infants with perinatal brain injury to consider the impact of the intervention on the population compared to standard neonatal intensive care. From an initial search identifying 7317 titles, 11 studies met the inclusion criteria and were included in this review. Four clinical trials (n = 2 MT interventions) had not published results and therefore were unable to be included in this review. The publication of these trials may contribute to the evidence base [56,57,58,59]. Additionally, there was one conference abstract [60] that exposed preterm infants to recordings of maternal sounds and voice four times in 24 h. Infants recruited included those with IVH grade I/II, with the intervention resulting in a reduction of time spent on respiratory support. This study was unable to be included in the results as the results were not published in a full article. Due to MT with neonates being an emerging field, most music interventions in included studies were not administered by a specialist music therapist. Interventions administered both by trained music therapists and healthcare professionals were included in this review, resulting in not all studies having a psycho-therapeutic approach. Studies considered the impact of MT, music exposure and vocal soothing upon infants with physiological and neurological parameters of brain injury as well as maternal experience.

### 4.1. The Effect of Auditory Stimulation on Infant Physiological and Neurological Parameters

Results from this review showed auditory stimulation, both pre-recorded and live, had positive effects on infant’s feeding ability, pain scores during procedures and neurological outcomes. Pre-recorded music was provided more often in a 24 h period than live interventions. When recordings are of maternal sounds/parental voice, this potentially provides parents with a sense of comfort in that their baby will still sense their presence even when they are not physically present. As with findings from other studies, our review identified that music interventions can help to reduce observed infant pain scores during painful procedures such as heel-lance and eye examinations [61]. Music in these studies was administered by healthcare professionals and could therefore be interpreted as auditory stimulation rather than having a therapeutic approach. Further studies into the impact of recorded music as a therapeutic intervention should consider the impact of pre-recorded music when created and administered with the assistance of a specialised music therapist with specialist neonatal training. These studies could then consider the potential for recorded, parent-informed music interventions to limit the negative impact of stress and protect the infant’s processing of environmental stimuli and social–emotional development whilst having a holistic approach.

Many of the studies included in the review commented on the state of the infant prior to the intervention. In Sa de Almeida et al.’s study, although music was pre-recorded, the track was selected based on the state of the infant. This study reported larger amygdala volumes as a result of musical intervention. It has been demonstrated that observing the state of an infant in order to adjust stimulation appropriately can have significant implications on premature infants’ neurodevelopment [62]. This concept informed the interaction between the music therapist and infant in Emery et al.’s study. Results of the developmentally focused sessions demonstrated MT’s ability to support infants with prolonged hospital care in the attainment of developmental milestones [52]. Ren et al. 2021 [63] used near-infrared spectroscopy to evaluate the impact of short-term music exposure on brain functions using pre-recorded music by Mozart. The authors found that whilst preterm infants were able to process the complex music, there were no significant changes in cerebral functional connectivity. However, when creative live MT was provided to premature infants (<32 weeks gestation at birth) by Haslbeck 2020 [24] and imaging data collected via resting-state MRI, an increased clustering coefficient and centrality was revealed. This provides evidence that live created music has a beneficial effect on functional brain connectivity in preterm infants. This is likely due to the music therapist’s ability to create music that is reflective of the infant’s state and behavioural responses. It would appear that whilst pre-recorded interventions have immediate behavioural responses such as reductions in pain scores [36], sleep cycles [64] and synchronization of sucking [41], neurological benefits are most likely with live music that is infant-directed. Two studies provided evidence for supporting parent education in order for the benefits of interventions to be continued after discharge. Further longitudinal research in which parents are encouraged and supported to continue the intervention post-discharge is required to determine any potential long-term impact of MT for premature infants, in particular for those with perinatal brain injury.

The majority of studies included in this review provided the intervention from 33 weeks postmenstrual age, with many of the musical interventions being complex—in particular, the use of recorded instrumental music. This methodology is supported by evidence that suggests the processing of complex sounds such as lullabies and stimulation of cortical activity occurs most significantly at around 33 weeks gestation [65,66]. However, it has been shown that foetuses demonstrate behavioural responses to sound from as young as 25 weeks and have a memory of sounds heard in utero into infancy [41,67]. This supports the use of live maternal voices from as young as 23 weeks in studies such as Letzkus et al.’s study, yet studies that included infants with PVL showed stress responses to the intervention. Future studies may consider a slow-paced introduction to stimulation for these infants which is then gradually increased. Additionally, ensuring the infant remains in their current behavioural state at the time of the intervention may aid the infant in processing the stimulation. A similar study [22] to the multi-stimulation studies included in the review was undertaken using MT as part of a multi-sensory bundle with neuro-typical premature infants from 30–32 weeks postmenstrual age. In the MT-led sensory bundle, authors suggested that this intervention supports infants’ toleration of stimulation and progression towards homeostasis. However, the study found positive effects on female infants only. The sex of the infants in the studies reviewed was not considered separately but is recognised as important as recent evidences indicate that neuronal cell death following brain injury is sex-dependent and mediated through different pathways [68].

Research demonstrates the benefit of early vocal interaction with neuro-typical infants for creating early dyadic synchrony and enhancing communication skills [69]. Additionally, research suggests that when engaged in musical exchange with shared rhythm and affects, the infant’s coregulation and stabilisation is supported [19,70]. By supporting early forms of social interaction, there is the potential to enhance the developmental outcomes of infants susceptible to socioemotional developmental delay; music has the ability to activate responses in brain regions associated with inducing emotions and regulation [71]. Whilst previous studies have often excluded those with severe brain injury, potentially due to the complexity of care these infants require and the possibility of further injury if hypoxic events were to occur, this warrants further exploration to determine potential long-term benefits to infants.

When children age 2 were compared using the Bayley Scales of Infant and Early Childhood development [47] after being placed as preterm infants (<30 weeks gestation) in private rooms or on an open ward for the duration of their stay, those who had been isolated had lower language (*p* = 0.006) and motor scores (*p* = 0.02) [72]. It is therefore necessary to conduct further research to consider ways of supporting infants diagnosed with perinatal brain injury to remain stable whilst accessing MT in order to support their long-term developmental outcomes. The observation of behavioural response and systemic variables can be used to guide interventions and increase potential benefits, ensuring that interventions are carried out safely. The support of professionals in neonatal observation can support parents with interpreting cues and model ways of using the voice to support the infant’s development. Specially trained music therapists can empower parents to use their voice to share family identities with their infant [43,73].

### 4.2. The Effect of Infant Auditory Stimulation upon Parental Experience

The findings from this study indicate that participation in auditory stimulation interventions is feasible for parents of infants with perinatal brain injury but can potentially result in an increase in anxiety. This is contradictory to current evidence on the impact of MT on parental well-being. There are several studies which have documented the positive effect of MT on the well-being and mental health of parents of preterm infants [74,75]. This includes parents of preterm neonates experiencing a reduction of stress and anxiety levels as result of participating in MT during kangaroo care [76]. When preterm infants with severe brain injury were included in Epstein et al.’s study, there was an increase in maternal anxiety recorded. There is evidence across a range of infant populations that MT reduces parental anxiety [23]. Kobus et al., 2021 [77] provided MT for infants in the state they were on approaching the infant and their family: either remaining in their cot or continuing kangaroo care with their parents (timings of sessions were arranged with parents and nurses). The authors found 81% of parents with preterm infants <32 weeks postmenstrual age stated that they were able to relax during MT, with 91% feeling that MT had improved the quality of their stay on the neonatal unit. It would therefore appear that the musical intervention itself could be unlikely to be the reason for the increase in anxiety in Epstein et al.’s study. Research investigating the use of live MT combined with kangaroo care with preterm and very preterm infants (median age 26 weeks post-menstrual age) suggests waiting until this population is >7 days of life before combining live MT with kangaroo care [78], which Epstein’s study followed. Van Dokkum et al., 2021 suggest that stimulation levels should be gradually increased starting with kangaroo care and then introducing voice with a Remo Ocean drum before later moving to voice and guitar after three sessions. A gradual approach has been used in other studies with preterm infants [24,64]. This suggests that the immediate use of lullabies accompanied by another voice or guitar in Epstein et al.’s study may have created a heightened alert state and increased maternal anxiety. Future studies including infants with brain injury should consider a similar gradual increase of stimulation. Infants with perinatal brain injury are likely to have additional monitoring as well as requiring minimal vestibular stimulation and therefore would require the support of a professional to ensure transfer to skin to skin was conducted safely. Further investigation is required into the appropriateness of providing MT interventions during the cooling period for infants who have experienced an HIE event and the acceptability of parents of this.

There is potential for live MT to be provided without additional interventions as a means of supporting with parental mental health and parent–infant interaction and to have a calming effect on infants in the period that minimal handling is advised before later combining with kangaroo care. It could be suggested that maternal singing in the presence of a music therapist resulted in an increase of stress for parents of infants with brain injury. The results of this study are not a representation of MT as an intervention, as only three studies were carried out by a qualified music therapist. The psychosocial aspect of MT and the value of the partnership between parents and therapist can therefore not be analysed. The benefits of a supportive relationship has not been discussed in the articles. However, when the support of a specialist in “Family Nurture Intervention” was provided and education materials provided for ongoing support in Letzkus et al.’s study, a reduction in maternal anxiety was shown. Additionally, parents in Emery et al.’s study were encouraged to hold their infants during delivery of MT and were educated on how to deliver the intervention themselves to continue the positive effects of the study. Emery et al. showed that the intensity of the frequency of sessions had no impact on the developmental outcomes of infants, enabling therapists to arrange sessions when parents are available without impacting outcomes. Future studies may consider the impact of the therapeutic relationship between professional and parent on parental experience and the long-term neurological outcomes of infants. Additionally, studies may consider the value of empowering parents and developing their confidence in their parenting abilities in order for parents to continue to understand how to support their infant’s development.

It is likely that cues of infants with complex medical needs and brain injury are more likely to be challenging to interpret than those of neurotypical and full-term infants. The engagement in MT can support with interpreting cues and responses and consequently support parent–infant interaction. This theory has resulted in the creation of the successful parent education program *Time Together* [79]. It could be suggested that, due to the increased challenges faced by parents of infants diagnosed with perinatal brain injury, this population would benefit from a similar program with heightened awareness of the appropriate stimulation levels and ways of supporting their infant’s development post-discharge. Although studies in this review were administered by healthcare professionals as well as music therapists, there is little indication of the professional’s understanding of the importance of parent–infant interaction. Future studies may consider the long-term impact of engaging cot-side nurses and other neonatal specialists in the understanding and provision of MT to continue to support and follow-up parent–infant interaction when in the unit and post-discharge. None of the studies reviewed evaluated the impact of the intervention on parent–infant bonding. The bonding process is known to be impacted by the trauma of premature birth and the management of infants with brain injuries. There is currently an on-going longitudinal study that aims to provide an evaluation of the effect of MT on parent–infant bonding with premature infants [80].

## 5. Limitations

As the aim of this study was to consider the impact of MT on infants with perinatal brain injury on the NICU, the number of studies meeting the eligibility criteria for this review was small. However, the studies identified highlighted a high number of RCTs providing evidence from a range of forms of intervention and focusing on different outcomes. Our limitation of the definition of MT being restricted to live vocal stimulation or pre-recorded music rather than including white noise also reduced the number of studies; however, it allowed us to include studies that also focused on parental outcomes, which are integral to family integrated care on a neonatal unit. In addition, for the purpose of this review, the principal of “communicative musicality” [81] was felt to be vital in determining the intervention and therefore felt to be present in vocal soothing but not white noise.

There were minimal studies to review, which created a limitation on the possibility of assessing the impact of different forms of auditory stimulation on infants with brain injury. Of these, only two considered the impact on parent experience, and therefore the results cannot offer a true representation of the impact. Whilst all studies that included any degree of newborn brain injury were included, some of these studies included only a small percentage of these infants in their total population and therefore did not discuss the results on those with brain injury separately. It was not possible to separate infants with perinatal brain injury, and therefore authors were unable to consider the impact of MT on these infants alone. It is therefore difficult to assess the true impact of the intervention on infants with brain injury. Additionally, the population of infants with brain injury often consisted of a high percentage of lower grades of injury, with no consideration of the varying developmental needs of the infants. Due to the minimal number of studies, the authors were unable to consider the impact of MT on specific pathologies.

## 6. Conclusions

This review found benefits of music interventions in both recorded and live forms for infants with perinatal brain injury. Musical interventions were shown to improve infant’s physiological parameters, particularly in the management of pain and feeding ability, and potential positive neurological outcomes are suggested with continued support post-discharge. Recorded maternal voices provided opportunities for parents to be connected to their infants when they were unable to be present with positive results. Infants diagnosed with PVL showed stress responses to some of the interventions provided with a prolonged alert state after the intervention was complete, demonstrating that a gradual approach to stimulation is required for this population alongside the careful monitoring of behavioural and physiological responses. The state of their infant influences a parent’s perception of an intervention, and therefore future studies should consider the support required for both parents and infants for the intervention to be advantageous. Neonatal trained music therapists can support the interpretation and understanding of infant cues so that stimulation can be adjusted appropriately. Further studies with this population using the assistance of a neonatal music therapist to support the provision of music are required to determine the appropriateness of MT for infants with perinatal brain injury and outcome measurements. These studies should include all birth parents and their partners for a wider understanding of the outcomes on parent experiences. Additionally, it is likely that this population would benefit from ongoing intervention post discharge for long-term neurological change due to the high neurodevelopmental needs. Future studies should consider support for families post-discharge. Large-scale randomised control trials would provide the scientific evidence-based research that MT with this population requires to become an acceptable complimentary approach or alternative to current medical treatment and to be incorporated into neonatal developmental care.

## Figures and Tables

**Figure 1 brainsci-12-00578-f001:**
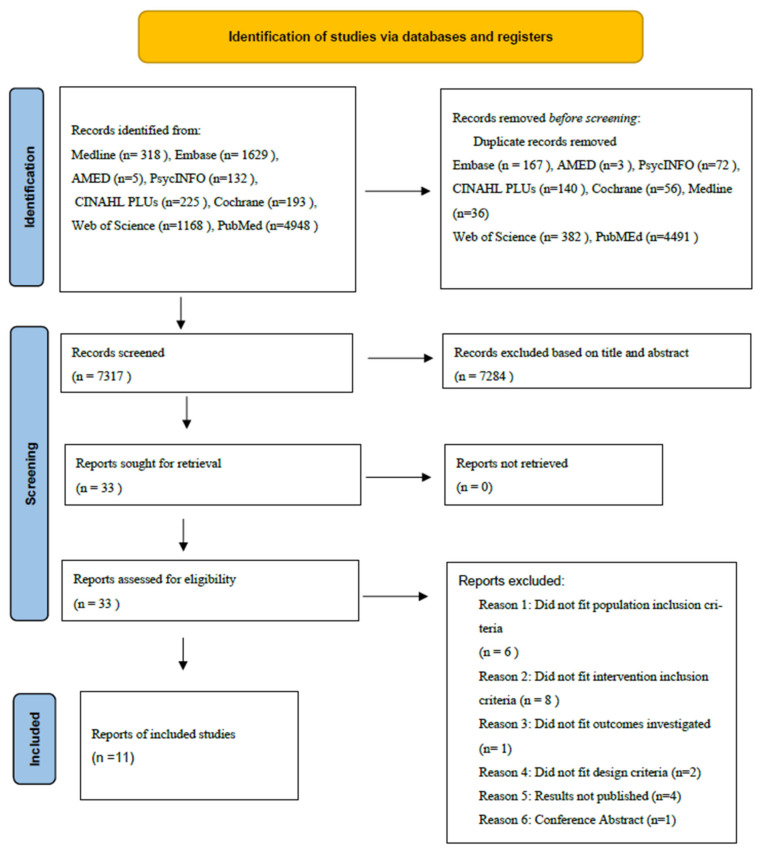
PRISMA flow chart of article retrieval.

**Table 1 brainsci-12-00578-t001:** Search strategy.

Database	Search Strategy
**PubMed**	(((((((((((music* adj3 therap*)) OR ((music intervention* or music or voice or song* or sound* or vocal or singing or womb sound* or heartbeat or heart beat))) OR ((music therap* or music stimulation or acoustic therap* or acoustic stimulation or auditory stimulation or auditory therap*))) OR (singing[MeSH Terms])) OR (sound[MeSH Terms])) OR (voice[MeSH Terms])) OR ((music[MeSH Terms]))) OR (Acoustic Stimulation[MeSH Terms])) OR (Music Therapy[MeSH Terms])) AND ((infant, newborn/or infant, premature/or infant, extremely premature/[MeSH Terms]) OR ((infant or neonat* or newborn* or new-born* or preterm or pre-term or extreme* preterm or extreme* premature or very preterm or moderate to late preterm or premature)))) AND ((Brain Injuries/or Ischemic Stroke/or Stroke/or Seizures/or Brain Injuries, Traumatic/[MeSH Terms]) OR ((brain injur* or seizure* or stoke* or neonatal encephalopathy or traumatic brain injur* or hypoxic ischemic encephalopathy or HIE or intrapartum asphyxia or f?etal brain injur* or IVH or intraventricular haemorrhage or PVL or periventricular leukomalacia))) Filters: Humans

**Table 2 brainsci-12-00578-t002:** Inclusion criteria.

**Population**	Neonates diagnosed with any form of brain injury
**Intervention**	Any music or vocal stimulation to both infant alone and with parent present.
**Comparator**	Form of auditory stimulation Frequency and duration of intervention Music therapy alone or combined with a second intervention Gestational age
**Outcomes**	Infants: MRI results Change in oxygen saturation Change in heart rate Change in behavioural state of infant Adverse effects: physiological signs of stress Weight gain Neurodevelopment Parents: Parent–infant bonding and attachment Parental mental health Change in level of anxiety
**Setting**	Neonatal Unit

**Table 3 brainsci-12-00578-t003:** Characteristics of included studies (VLBW: very low birth weight; NT: neurotypical; PVL: periventricular leukomalacia; IVH: interventricular haemorrhage; FT: full term; PTM: preterm music; PTC: preterm control).

First Author, Year	Country	Number of Participants	Study Design	Gestation at Birth (Weeks)	Postconceptional Age at Time of Intervention	Form of Brain Injury	Intervention: From of Stimulation Received/ Lead Administrator of Invention	Duration per Session/Occurrence Study Duration	Outcomes Measured	Results
Chirico, G., 2017 [36]	Italy	40	RCT	Group 1: 29.4 ± 2.6 Control: 30.8 ± 2.4 (Mean age)	Group 1: 33.4 ± 2.0 Control: 33.3 ± 1.9 (Mean wks.)	Includes those with IVH grade I &II (percentage of population unspecified)	Exposure to recordings of mother’s voice (lullabies and nursery rhymes) Healthcare professional led	10 min 2 times daily Study duration: 3 days	Premature infant pain profile, heart rate, oxygen saturation, blood pressure, side effects: apnoea, bradycardia, seizures and vomiting.	Infants in treatment group had lower PIPP scores (*p* = 0.00002) and lower decrease in oxygen saturations (*p* = 0.0283). No significant side effects were observed.
Chorna, O.D., 2014 [35]	USA	94	RCT	Median: Group 1: 30 Control: 30	Percentage of 34 completed weeks: Group 1: 93% Control: 98%	White matter injury (all types): Intervention 17% Control 19% White matter injury, severe (PVL/IVH grade III and IV): Intervention 4% Control 6%	Pre-recorded mothers singing on Pacifier Activated Music player (PAM) Music therapist support for recording of mother’s voices. Intervention the activated by infant’s suck and monitored by nursing staff.	15 min daily Study duration: 5 days	Feeding rate, suck pressure, number of completed feeds, weight on discharge	Musical intervention increased oral feeding rate (2.0 vs. 0.9 mL/min *p* < 0.001), oral volume intake (91.1 vs. 48.1 mL/kg/d, *p* = 0.001) oral feeds/day (6.5 vs 4.0, *p* < 0.001) and faster time to full oral feedings (31 vs. 28 d, *p* = 0.04) compared to controls.
Doheny, L., 2012 [37]	USA	14	Cohort	Mean 30.2 ± 2.07	Mean wks.: 31.1 ± 2.06	7.1% with Grade III or IV IVH	Exposure to pre-recorded maternal voice with heartbeat Healthcare professional led. Nursing staff administered sounds avoiding parent visits and clinical exams.	Four sessions of 30 min for each observed 24 h period per week Duration of study: 24 hperiod observed for a mean period of 42.9 ± 25.07 days.	Frequency of cardiorespiratory events (CRE)	Decrease in trend in CRE with age. With maternal sounds a lower frequency of CRE were observed. This was most evident in infants ≥33 weeks postmenstrual age (*p* = 0.03).
Emery, L., 2018 [52]	USA	24	RCT	Median 28.5	49.5 weeks	IVh grade I–IV 13%	Developmental music therapy Music therapist led	Group A: 2 sessions per week for 4 weeks, a minimum of one day of no therapy between sessions Group B: 4 sessions for weeks 1 and 3, no sessions on weeks 2 and 4 Duration of study: 4 weeks	Comparison of intensive versus standard spaced protocolised music therapy on developmental milestone acquisition	Developmental MT supports developmental skill acquisition. Intensity of intervention had no effect on the degree of skill attained.
Epstein, S., 2020 [44]	Germany	35	RCT	Mean: 27 ± 2.5	Mean wks.: Group 1: 32 ± 3.5 Control: 31 ± 2.7	IVH grade III 45% IVH grade IV 29%, PVL 26%	Maternal singing alone compared to singing combined with skin to skin Music therapist led	Three sessions 20 min Duration of study time (days): Skin to skin group 18 ± 3.2 Skin to skin with MT 20 ± 2.1	Oxygen Saturation, Respiratory rate, behavioural state, mothers’ anxiety (STAI score)	Skin to skin with maternal singing: Higher mean ± standard deviation (SD) LF/HF ratio (1.8 ± 0.7 vs. 1.1 ± 0.25, *p* = 0.01), higher mean ± SD heart rate (145 ± 15 vs. 132 ± 12 beats per minute, *p* = 0.04), higher median infant behaviour state (NIDCAP manual for naturalistic observation and the Brazelton Neonatal Behavioural Assessment) score (3 (2–5) vs. 1 (1–3), *p* = 0.03) and higher mean ± SD maternal anxiety (state-trait anxiety inventory) score (39.1 ± 10.4 vs. 31.5 ± 7.3, *p* = 0.04)
Konar, M.C., 2021 [38]	India	3095	RCT	Group A: 34.3 ± 3.1 Group B: 34.5 ± 3.0	Not stated	Stages of HIE Experimental group: I 50.8% II 34.5% III 14.7% Control group: I 50.2% II 34.7% III 11.6%	Exposure to recordings of Rabindra Sangeet (guitar) Healthcare professional led on the unit/caregiver led after discharge.	4 sessions daily 25–30 min Duration of study time: On the unit: Experimental group 13.1 ± 2.3 Control 16 ± 2.1 Followed up after discharge at 3, 6, 12, 18, and 24 months.	Hospital stay, oxygen dependency, refractory convulsion, apnoea, cumbersome method, pain score (N-PASS), motor neurodevelopmental quotient (DASII)	Mean hospital stay, oxygen dependency, requirement of mechanical ventilation and frequency of apnoea and pain score were lower with music intervention. Additionally, the group receiving musical intervention showed better neurodevelopmental results at all stages (3, 6, 12, 18 and 24 months)
Letzkus, L., 2021 [53]	USA	11	Feasibility	Median 27 ±2.85 ± 2.85	Different stages of bundle started at different ages: 23–25 week- Vocal soothing, Scent exchange, comforting touch 26–28 weeks: addition of kangaroo care 29–32 weeks: infant massage when deemed appropriate	IVH 58.3% (Total n = 7: unilateral IVH n = 4, bilateral IVH n = 3. Grade I/II n = 6, Grade II/III n = 1) Cystic periventricular leukomalacia 16.6%	Maternal vocal soothing as part of NICU-bundle (vocal soothing, scent exchange, comforting touch) Healthcare professional with familiarity of the Family Nuture Intervention led. Additional education in the form of intervention and method flip book.	Minimum 5 days per week. Duration of study: Median 8 weeks	Feasibility of carrying out bundle 5 days/week, Maternal stress and anxiety, Short-term Motor outcomes (GMA/HINE)	Vocal soothing, scent exchange and comforting touch were performed at or above the predetermined goal of 71% of time (5/7 days), kangaroo care and infant massage were not. Decrease in maternal stress, anxiety and depression during the study.
Nelson, M.N., 2001 [54]	USA	37	RCT	Mean VLBW Experimental: 25.45 ± 1.13 Control: 25.60 ± 1.52 PVL: Experimental: 27.20 ± 2.82 Control: 27.27 ± 2.37	Mean VLBW Experimental: 33.64 ± 0.5 Control: 33.60 ± 0.98 PVL: Experimental: 33.40 ±1.07 Control: 33.64 ± 0.92	IVH (grade III/IV) Or IVH (grade III/IV) with PVL Or PVL alone.	Spoken female voice as part of auditory-tactile- visual-vestibular Intervention Caregiver led.	Two sessions daily/5 days per week 15 min 70% of infants completed the intervention for 12 months	One year developmental outcomes Mother-infant interactions (Dyadic Mutuality Code and Nursing Feeding assessment scale)	PVL was associated with poorer development regardless of group assignment. Experimental infants had 23% fewer cerebral palsy diagnoses at 1 year. No significant difference between VLBW group and PVL group in early mother-infant interaction.
Sa de Almeida, 2020 [46]	Switzerland	45	RCT	Mean: FT: 39.2 ± 1.3 PTM: 28.58 ± 2.3 PTC: 28.30 ± 2.3	From 33 weeks until discharge Age at MRI: FT: 39.54 PTM: 40.15 ± 0.6 PTC: 40.48 ± 0.6	IVH grade I and II included (Percentage) FT: 0 PTM: 26.7% PTC: 6.7%	Exposure to music by Vollenweider (instrumental) through headphones Healthcare professional led.	Eight minutes for a mean 4.84 ± 1.18 per week Total duration of days on study unspecified.	White matter maturation, T2-weighted image (amygdala volumetric analysis)	Improvement in white-matter maturation in acoustic regions, external capsule/claustrum/extreme capsule and uncinate fasciculus as well as larger amygdala volumes in preterm infants exposed to music intervention.
White-Traut, R., 1999 [55]	USA	30	RCT	Mean wks. Control: 29.7 ± 2.06 Experimental: 27.9 ± 2.33	Mean wks. Control: 33.7 ± 0.59 Experimental: 33.5 ± 0.74	PVL Acute/Cystic Control: 6/9 Experimental 9/6	Female voice as part of multisensory intervention (auditory, tactile, visual and vestibular) Research assistant led	Fifteen minutes twice a day, 5 days a week Duration of study: 4 weeks	Neurobehavior (Brazelton neonatal behavioural assessment scale) and neurodevelopment (Bayley scales of infant development)	Experimental group demonstrated a significant shift from sleep to alert during the intervention. No injury was sustained by the experimental group. Average hospital stay of the experimental group was 9 days shorter than controls.
White-Traut, R., 2003 [45]	USA	37	RCT	Mean wks. Group 1: 26.29 ± 2.239 Control: 26.75 ± 2.236	Mean wks. Group 1: 33.524 ± 0.814 Control: 33.625 ± 0.885	PVL/IVH grade III/IV/PVL&IVH grade III/IV	Female voice as part of multisensory intervention (voice with eye contact and stroking followed by rocking) Research assistant led	Twice daily until discharge 15 min in total per intervention Observed for 2 weeks (33–35 postconceptional age)	Heart rate, respiratory rate, haemoglobin oxygen saturation	Those without central nervous system injury demonstrated a decrease in resting mean heart rate with stable respiratory rate and oxygen saturation. Infants with PVL showed increase in heart rate post-intervention.

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
