# Peer review of "The Role of Music Therapy with Infants with Perinatal Brain Injury"

_brainsci, 2022, doi:10.3390/brainsci12050578_

Round 1

Reviewer 1 Report

The Role of Music Therapy with Infants with Perinatal Brain Injury

By Ormston et al

The authors performed a systematic review of evidence on the use of music therapy for infants with perinatal brain injury, including hypoxic-ischemic encephalopathy in fullterm infants and intraventricular hemorrhages or periventricular leukomalacia in preterm infants. Even though evidence seems limited, this review contributes to the emerging field of music therapy in neonatal care. For infants with brain injury, specific interventions may be designed based on the findings of this review. The manuscript is well written and addresses an important topic. I have the following comments and questions:

The introduction is well written and clearly structured. Only a few minor issues:

  • Line 57 misses a full stop at the end of the sentence.
  • Line 87 should say ‘neonatal units’ instead of ‘neonatals unit’.

Methods:

  • The introduction and aim clearly state looking at preterm infants and their brain injuries as well, but these terms are lacking from the search strategy, i.e. terms such as intraventricular hemorrhages or periventricular leukomalacia are not included. Did the authors miss any publications without these terms in their search strategy?
  • The Figure on page 5 does not show the full text box on excluded reports. Can the authors align so that readers can see all?

Results:

  • The distinction between music in medicine and music therapy does not seem to be made based on the inclusion criteria for the included articles, in studies with pre-recorded music, was a music therapist involved? If not, was it truly music therapy? Authors may want to consider calling this pre-recorded music instead of pre-recorded music therapy.
  • In studies where infants with and without perinatal brain injury were included and analyzed together, were the authors able to disentangle the group with perinatal brain injury from other infants, to specifically study the effect of music therapy on this population alone?
  • From Table 3, it is obvious that the grade of IVH or PVL is not always known or described. This may have implications for the music therapy intervention and its effect on the neonate. Did the authors consider this? Do the studies report when the music therapy or pre-recorded music was played and what the grade of IVH/PVL was at that time? Were any interventions provided during active cooling therapy for the HIE babies?
  • Line 369-370 has a striked-through word ‘intervention’ that should be removed.

The discussion is well written and addresses all controversies in the currently emerging field of music therapy in neonatal care.

Author Response

Thank you for your comments. We have re-run the searches including IVH/ PVL and included the articles which met criteria. The article has been formatted in UK English and table adjusted accordingly. 
Due to music therapy being an emerging field there are very few music therapist led articles published and so we have included those with infants with any degree of brain injury. Due to the minimal number of results we were unable to consider the impact on varying degrees of injury. We have made edits to the article to reflect the difference between music therapist led sessions and music exposure and further explain the gaps in current literature. We hope this improves the quality of the review. 

Reviewer 2 Report

This is an interesting review about the interest of music intervention in infants with perinatal brain injury. the manuscript was clear and well written. However, I have major issues with the data interpretation based on the results that would need to be addressed before the manuscript is reconsidered.
Major comments: - the authors found only nine studies among which seven randomized controlled trials and only 4 were RCTs using music/singing as the intervention. The others included auditory stimulation which is different than music therapy (2 included multisensory intervention). It is not clear why such studies are included in a systematic review on music therapy. Also, consequently, in the discussion, the authors should highly moderate any advice regarding such intervention since it is only based on 4 studies who included very different types of brain injuries, frequency of intervention, and outcome measures. Adopting a descriptive approach highlighting the gaps and future research paths would be more appropriate. Minor comments: - statistical results should not be included in the abstract as it can be confusing for the reader as this being metanalyses results (while these are only results reported from original articles) - in the table, the overall duration of the intervention is not mentioned for most studies (1week,2weeks,...)  

Author Response

Thank you for your comments, we have found them very useful. We have re-run searches to include terms IVH/ PVL which has resulted in two more articles being included. As music therapy on the neonatal unit is an emerging field we included any form of intervention which was felt to have a musical content. This has included vocal soothing due to the musical manner which parents/ caregivers interact with infants. Decisions for inclusion criteria have been made and added to the methods. We have edited the review to bring clarity to the difference between Music Therapy interventions and music exposure and highlight the need for music therapy specific research. Unfortunately due to the limited number of studies we were unable to consider the impact of music interventions on specific degrees of brain injury. We have made edits to the article to reflect this and changed the discussion to highlight areas for future research as suggested. 

Results have been removed from the abstract and the table now includes: 

Administrator of intervention

Duration of study

Degree of injury and percentage of population

We hope that these edits improve the article and address your suggestions.

Round 2

Reviewer 2 Report

The manuscript has substantially been revised to address the comments made.

The current version is improved and suitable for publication in brain sciences.